# Lifelong Word Embedding via Meta-Learning

## Abstract

Learning high-quality word embeddings is of significant importance in achieving better performance in many down-stream learning tasks. On one hand, traditional word embeddings are trained on a large scale corpus for general-purpose tasks, which are often sub-optimal for many domain-specific tasks. On the other hand, many domain-specific tasks do not have a large enough domain corpus to obtain high-quality embeddings. We observe that domains are not isolated and a small domain corpus can leverage the learned knowledge from many past domains to augment that corpus in order to generate high-quality embeddings. In this paper, we formulate the learning of word embeddings as a lifelong learning process. Given knowledge learned from many previous domains and a small new domain corpus, the proposed method can effectively generate new domain embeddings by leveraging a simple but effective algorithm and a meta-learner, where the meta-learner is able to provide word context similarity information at the domain-level. Experimental results demonstrate that the proposed method can effectively learn new domain embeddings from a small corpus and past domain knowledges[1]. We also demonstrate that general-purpose embeddings trained from a large scale corpus are sub-optimal in domain-specific tasks.

## 1 Introduction

Learning word embeddings (Mnih & Hinton (2007); Turian et al. (2010); Mikolov et al. (2013a;b;c); Pennington et al. (2014)) has received a significant amount of attention due to its high performance on many down-stream learning tasks. Word embeddings have been shown effective in NLP tasks such as named entity recognition (Sienčnik (2015)), sentiment analysis (Maas et al. (2011)) and syntactic parsing (Durrett & Klein (2015)). Such embeddings are shown to effectively capture syntactic and semantic level information associated with a given word (Mikolov et al. (2013a)).

The "secret sauce" of training word embedding is to turn a large scale in-domain corpus into billions of training examples. There are two common assumptions for training word embeddings: 1) the training corpus is largely available and bigger than the training data of the potential down-stream learning tasks; and 2) the topic of the training corpus is closely related to the topic of the down-stream learning tasks. However, real-world learning tasks often do not meet one of these assumptions. For example, a domain-specific corpus that is closely related to a down-stream learning task may often be of limited size. If we lump different domain corpora together and train general-purpose embeddings over a large scale corpus (e.g., GloVe embeddings (Pennington et al. (2014)) are trained from the corpus Common Crawl, which covers almost any topic on the web), the performance of such embeddings on many domain-specific tasks is sub-optimal (we show this in Section 6). A possible explanation is that although many domain words share similar meanings with the same out-of-domain words, with no in-domain awareness, dumping many out-of-domain co-occurrences as training examples may bias in-domain embeddings. (e.g., if the domain is about food, then an out-of-domain "python" as a programming language can bias "java", while the in-domain word "chocolate" is more likely to help).

To solve the problem of the limited domain corpus, one possible solution is to use transfer learning (Pan & Yang (2010)) for training domain-specific embeddings (Bollegala et al. (2015); Yang et al. (2017)). However, these methods just manage to leverage out-of-domain embeddings trained from a large scale corpus to help limited in-domain corpus. The very in-domain corpus is never expanded.

---

[1]We will release the code after final revisions.

Also, one common assumption of these works is that a pair of similar source domain and target domain is manually identified in advance. In reality, given many domains, manually catching useful information in so many domains are very hard. In contrast, we humans learn the meaning of a word more smartly. We accumulate different domain contexts for the same word. When a new learning task comes, we may quickly identify the new domain contexts and borrow the word meanings from existing domain contexts.

This is where lifelong learning comes to the rescue. Lifelong machine learning (LML) is a continual learning paradigm that retains the knowledge learned in past tasks $1, \ldots, n$, and uses it to help learning the new task $n + 1$ (Thrun (1996); Silver et al. (2013); Chen & Liu (2016)). In the setting of word embedding: we assume that the learning system has seen $n$ domain corpora: $(D_1, \ldots, D_n)$, when a new domain corpus $D_{n+1}$ comes by demands from that domain's potential down-stream learning tasks, the learning system can automatically generate word embeddings for the $n + 1$-th domain by effectively leveraging useful past domain knowledge.

The main challenges of this task are 2 fold. 1) How to identify useful past domain knowledge to train the embeddings for the new domain. 2) How to *automatically* identify such kind of information, without help from human beings. To tackle these challenges, the system has to learn how to identify similar words in other domains for a given word in a new domain. This, in general, belongs to meta-learning (Vilalta & Drissi (2002); Peng et al. (2002)). Here we do not focus on specific embedding learning but focus on learning how to characterize corpora of different domains for embedding purpose.

The main contributions of this paper can be summarized as follows: 1) we propose the problem of lifelong word embedding, which may benefit many down-stream learning tasks. We are not aware of any existing work on word embedding using lifelong learning 2) we propose a lifelong embedding learning method, which leverages meta-learning to aggregate useful knowledge from past domain corpora to generate embeddings for the new domain.

## 2 RELATED WORKS

Learning word embeddings has been studied for a long time (Mnih & Hinton (2007)). Many earlier methods employ complex neural network architectures (Collobert & Weston (2008); Mikolov et al. (2013c)). Recently, a simple and effective unsupervised model called skip-gram (Mikolov et al. (2013b;c)) was proposed to turn plain text corpus into large-scale training examples without any human annotation. It uses the current word to predict the surrounding words in a context window by maximizing the likelihood of the predictions. The learned parameters for each word are then the embeddings of that word. Although such embeddings can be trained in large scale and easily obtained online (Pennington et al. (2014); Bojanowski et al. (2016)), they are sub-optimal for many domain-specific tasks (Bollegala et al. (2015); Yang et al. (2017)). Domain corpus also suffers from limited size to train high-quality embeddings.

Our work is most related to Lifelong Machine Learning (LML)) (or lifelong learning). Much of the work on LML focused on supervised learning (Thrun (1996); Silver et al. (2013); Ruvolo & Eaton (2013); Chen & Liu (2016)) Recent years, several works have also been done in the unsupervised setting, mainly on topic modeling (Chen & Liu (2014)), information extraction (Mitchell et al. (2015)) and graph labeling (Shu et al. (2016)). However, we are not aware of any existing research that has been done on using lifelong learning for word embedding. LML is related to transfer learning and multi-task learning (Pan & Yang (2010)), which have been leveraged in word embeddings (Bollegala et al. (2015); Yang et al. (2017)). However, LML is different from transfer learning (see the survey book from Chen & Liu (2016)). Given many domains with uncertain relevance for the new domain, the lack of guidance on which kind of information is worth learning from the past domains is a problem. And there's no good measure of similarity of two words in different domains.

The proposed method leverages meta-learning (Vilalta & Drissi (2002)), which is to perform machine learning on learning tasks. Recently, meta-learning (or learning to learn) has been used to learn parameters of an optimizer (Andrychowicz et al. (2016)), to learn neural architectures (Fernando et al. (2017)). We leverage a meta-learner to accumulate knowledge during lifelong learning.

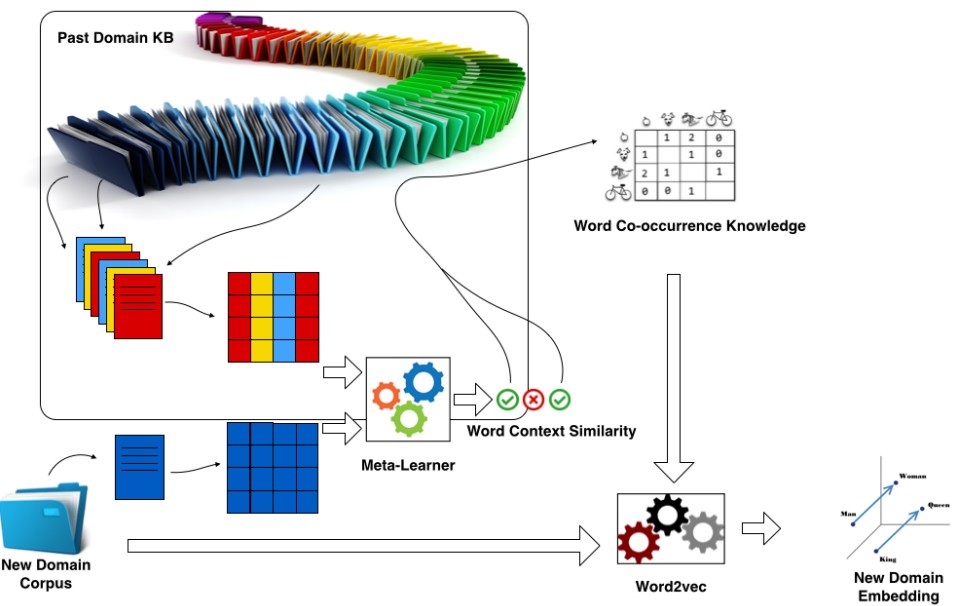

Figure 1: Lifelong word embedding learning process.

# 3 MODEL OVERVIEW

The overall lifelong learning process is depicted in Figure 1. Given a series of domain corpora $D_n = d_1, d_2, \ldots, d_n$, the lifelong learning system first learns a meta-model (learner) on domain-level word context similarity from the first $m$ domain corpus. As more domains arrive, the system accumulates the knowledge of domain corpora. When a new domain $D_{n+1}$ comes, the system uses the meta-learner to catch past domain knowledge that is useful and related to the new domain $D_{n+1}$ as augmented knowledge. With the augmented knowledge, the word embedding learning process is performed and the resulting embeddings are used for further down-stream learning tasks. The meta-learner here plays a central role in automatically identifying useful knowledge from past domains to help the new domain. By using a pairwise network, the meta-learner finds words in the past domains that are similar to the new domain. Then the co-occurrence knowledge of those similar words from the past domain is used together with the new domain corpus to train the new domain embeddings.

# 4 META-LEARNER

In this subsection, we describe how a meta-learner can help to identify similar words from many past domain corpora. When it comes to borrowing knowledge from past domains, the first problem is what to borrow. Although binary cross-domain embeddings are studied in Bollegala et al. (2015); Yang et al. (2017), they mostly assume that a relevant domain is already identified and shared words between two domains have similar meanings. In reality, given a wide spectrum of domains, borrowing knowledge from a non-relevant domain may not be helpful or even harmful to word embeddings (we show this Section 6). The meaning of one word in one domain may be quite different from the same word in another. For example, the word "java" in the programming context is different from the restaurant context. Borrowing the knowledge from a restaurant corpus can be harmful to the representations of "java" in a programming context.

## 4.1 TRAINING EXAMPLES

On top of learning embeddings for specific domains, we build a meta-learner to learn a general word context similarity from the first $m$ domains, where $m \ll n$. In practice, if $n$ is small, $m$ domains can simply be sampled from $n$ domains. Here since our experiments are conducted on hundreds of domains, we hold-out $m$ domains to train the meta-learner. The expected input to the meta-learner is

a pair of the same word from similar ("java" from two corpora of the restaurant domain) or different domains (e.g. "java" from the restaurant domain or the programming domain). The output of the meta-learner is whether two words are from the same domain or not. We first characterize words in domain corpora. Given a specific word in a domain, we choose its co-occurrence counts with $f$ frequent words within a context window (like word2vec) as the discrete features (a sparse vector of length $f$) of a word in that domain. This is inspired by the fact that a good dictionary (e.g. Longman dictionary) uses only a few thousand words to explain all other words. We denote the selected the top $f$ frequent words over $m$ domains as $V_{wf}$. Then given a domain corpus $D_i$, we sample $l$ sub-corpora $D_{i,j} \sim P(D_i)$ by selecting a fixed amount of chunks in $D_i$. A chunk can be a sentence or a document in $D_i$. We randomly select a fixed amount of chunks because the word features built from the sub-corpus are expected to be on the same scale. Then we randomly select a subset of words from top $f$ words as training example words $V_{meta}$. These training example words are the same in all domains $D_{1:m}$. We use these words in $V_{meta}$ as co-occurrence features and build features $\boldsymbol{u}_{w_{i,j,k}}$ for the word $w_k \in V_{meta}$ on the $j$-th sub-corpus of the $i$-th domain. We build word features for all $m$ domain sub-corpora $D_{1:m,1:l}$.

Finally, a pairwise meta-learner is trained on pairs of word features drawn from different domain sub-corpora for the same word. Given a word $w_k \in V_{meta}$, a pair of word features $(\boldsymbol{u}_{w_{i,j,k}}, \boldsymbol{u}_{w_{i,j',k}})$, where $j \neq j'$, forms a postive example; whereas $(\boldsymbol{u}_{w_{i,j,k}}, \boldsymbol{u}_{w_{i',j',k}})$ with $i \neq i'$ ($j$ and $j'$ can be equal or not) forms a negative example.

The $m$ domains are split into disjoint $m_t$ training domains, validation domains, and testing domains. So both the validation and testing examples are unseen examples during training. We enforce such isolation and wish the trained meta-learner can be more generally applied to the rest $n - m$ domains.

## 4.2 PAIRWISE NETWORK

We train a simple but efficient neural network to learn pairwise domain-level word context similarity. The idea of making such a network small but high-throughput is crucial in lifelong settings. This is because the meta-learner is heavily used in the later lifelong learning process. Given so many domains with so many words asking for detecting similarity, a small pairwise network with fewer parameters is desirable to save more memory being used for high-throughput inference.

The proposed pairwise network contains only one shared fully-connected layer (normalized by the co-occurrence feature) to learn continuous features from co-occurrence (discrete) features, a cosine function to learn similarity and a sigmoid layer to generate predictions like linear regression. The network is parameterized as follows:

$$\sigma(W_2 \cdot \text{Cosine}((\boldsymbol{W}_1 \cdot \frac{\boldsymbol{u}_{w_{i,j,k}}}{|\boldsymbol{u}_{w_{i,j,k}}|_1}), (\boldsymbol{W}_1 \cdot \frac{\boldsymbol{u}_{w_{i',j',k}}}{|\boldsymbol{u}_{w_{i',j',k}}|_1})) + b_2), \tag{1}$$

where $|\cdot|_1$ is the $l1$-norm, $\boldsymbol{W}$s and $b$ are weights and $\sigma(\cdot)$ is the sigmoid function. Cosine similarity is defined as $\text{Cosine}(\boldsymbol{x}, \boldsymbol{y}) = \frac{\boldsymbol{x} \cdot \boldsymbol{y}}{||\boldsymbol{x}||_2 \cdot ||\boldsymbol{y}||_2}$. Most trainable weights of this simple network reside in $\boldsymbol{W}_1$, which learn continuous features on the $f$ words. These weights can also be interpreted as an embedding matrix for the $f$ words. These $f$ word embeddings serve as general word embeddings to explain domain-specific words. We train the meta-learner over a hold-out domain set as the base meta-learner $M$. Then we fine-tune the meta-learner based on new domain corpus, as described in the next section.

## 5 LIFELONG LEARNING PROCESS

The previous section ends up with a base meta-learner $M$. In this section, we describe how the lifelong learning system works based on $M$, $n - m$ domains, and the new domain corpus $D_{n+1}$.

## 5.1 WORD CONTEXT RETRIEVAL

Assume the lifelong learning system has seen $n$ domain corpora. The system stores knowledge into a knowledge base $\mathcal{K}$. The knowledge base $\mathcal{K}$ contains a base meta-learner $M$ trained over the first $m$ domains, fine-tuned meta-learners $M_{m+1:n}$ and knowledge over past $n - m$ domains. The

knowledge includes the vocabulary of word features $V_{wf}$, $n - m$ domain corpora $D_{m:n}$, vocabularies on $n - m$ domains $V_{m:n}$, and word features on those vocabularies $E_{m:n}$. The word features $E_{m:n}$ are computed from one sample from each domain corpus.

Given a new domain corpus $D_{n+1}$, the lifelong learning system first fine-tunes the base meta-learner $M$. This ends with a fine-tuned meta-learner $M_{n+1}$ for this new domain. The tuning process makes the meta-learner more suitable for the new domain to retrieve past knowledge. Tuning examples are sampled similarly as the training examples of base meta-learner, except that negative examples are sampled between $D_{n+1}$ and $D_{m+1:n}$. Then the lifelong learning system retrieves similar in-domain word context information as augmented knowledge, which is used in embedding training in the next subsection.

The retrieval process is described in Algorithm 1. Firstly, line 1-2 build word features for the new domain corpus. These two operations are already done when preparing fine-tuning data for the meta-learner. Here we just mention them for storing knowledge purpose in line 13. Line 3 retrieves past domain knowledge, which is the reversed process similar to line 13 for the new domain. Line 4 defines the variable that stores useful past knowledge. Line 5-12 retrieves relevant words from past domains and store them in $\mathcal{A}$. More importantly, the fine-tuned meta-learner at line 9 finds similar words from past domains. Then we only keep similar words with a probability higher than a threshold $delta$ at line 10. This threshold controls the quality of the accumulated words $O$. Line 11 retrieves co-occurrence information about the set of words $O$ from past domain $D_i$ as augmented knowledge. Such augmented knowledge is more close to the co-occurrence information in the new domain $D_{n+1}$. ScanCooccurrence$(D_i, O)$ can be viewed as scanning the past domain corpus and grab the word in $O$ with its context words like word2vec, or as retrieving rows of co-occurrence counts from the co-occurrence matrix of the domain $D_i$. Line 13 simply stores the knowledge of the new domain for further use.

---

**Algorithm 1:** Lifelong domain-level word context retrieval

**Input**   : a knowledge base $\mathcal{K}$ containing knowledge over past $(n - m)$ domains,
             a new domain corpus $D_{n+1}$,
             and a fine-tuned meta-learner $M_{n+1}$.

**Output**: a word co-occurrence set $\mathcal{A}$, where each element is a 2-tuple $(w_t, w_c)$, representing useful
             knowledge from past domains.

1   $V_{n+1} \leftarrow$ BuildVocab$(D_{n+1})$
2   $E_{n+1} \leftarrow$ BuildWordFeature$(D_{n+1}, \mathcal{K}.V_{wf})$
3   $(D_{m:n}, V_{m:n}, E_{m:n}, M_{m:n}) \leftarrow \mathcal{K}_{m:n}$
4   $\mathcal{A} \leftarrow \emptyset$
5   **for** $(D_i, V_i, E_i) \in D_{m:n}, V_{m:n}, E_{m:n}$ **do**
6        $O \leftarrow V_i \cap V_{n+1}$
7        $F_i \leftarrow$ RetrieveWordFeature$(E_i, O)$
8        $F_{n+1} \leftarrow$ RetrieveWordFeature$(E_{n+1}, O)$
9        $S \leftarrow M_{n+1}$.inference$(F_i, F_{n+1})$
10      $O \leftarrow \{o | o \in O \text{ and } S[o] \geq \delta\}$
11      $\mathcal{A} \leftarrow \mathcal{A} \cup$ ScanCooccurrence$(D_i, O)$
12   **end**
13   $\mathcal{K}_{n+1} \leftarrow (D_{n+1}, V_{n+1}, E_{n+1}, M_{n+1})$
14   **return** $\mathcal{A}$

---

## 5.2   LIFELONG WORD EMBEDDING

In this subsection, we first describe the skip-gram model introduced by Mikolov et al. (2013b) in the context of a new domain in the lifelong setting. Given a new domain corpus $D_{n+1}$ with a vocabulary $V_{n+1}$, the goal of the skip-gram model is to learn a vector representation for each word $w \in V_{n+1}$ in that domain. Assume the domain corpus is represented as a sequence of words $D_{n+1} = (w_1, \ldots, w_T)$, the objective of the skip-gram model is to maximize the following log-

likelihood:

$$\mathcal{L}_{D_{n+1}} = \sum_{t=1}^{T} (\sum_{c \in \mathcal{C}_t} (\log \sigma(\boldsymbol{u}_{w_t}^T \cdot \boldsymbol{v}_{w_c}) + \sum_{c' \in \mathcal{N}_t} \log \sigma(-\boldsymbol{u}_{w_t}^T \cdot \boldsymbol{v}_{w_{c'}}))), \quad (2)$$

where $\mathcal{C}_t$ is the set of indices of words surrounding word $w_t$ in a fixed context window; $\mathcal{N}_t$ is a set of indices of words (negative samples) drawn from the vocabulary $V_{n+1}$ for the $t$-th word; $\boldsymbol{u}$ and $\boldsymbol{v}$ represent word vectors (or embeddings) we are trying to learn. The goal of skip-gram is to independently predict the presence (or absence) of context words $w_c$ given the word $w_t$. When the size $T$ of the corpus is extremely large, the skip-gram model, in fact, can be fed with billions of training examples. So the vector of a word can be trained to have a good representation of the similarity with the word's context words.

However, depending on the specific down-stream tasks, many domain corpora may not have a large scale corpus. And a random sequence of words drawn from other domains may not truly reflect the distribution $P(w_c|w_t)$ in domain $D_{n+1}$. This is where the previously computed augmented word co-occurrence $\mathcal{A}$ come to rescue. Assume our lifelong learning system has seen $m$ domains to build the meta-learner $M$ and $n - m$ domains to build the knowledge into $\mathcal{K}$. Given a new domain corpus $D_{n+1}$, we first perform Algorithm 1 to obtain the augmented word co-occurrence from past domains $\mathcal{A}$. Then this co-occurence information $\mathcal{A}$ is integrated into the objective function of skip-gram as following:

$$\mathcal{L}'_{D_{n+1}} = \mathcal{L}_{D_{n+1}} + \sum_{(w_t, w_c) \in \mathcal{A}} (\log \sigma(\boldsymbol{u}_{w_t}^T \cdot \boldsymbol{v}_{w_c}) + \log \sigma(-\boldsymbol{u}_{w_t}^T \cdot \boldsymbol{v}_{w_{c'}})), \quad (3)$$

where $w_{c'}$ is a random word drawn from the vocabulary. We use the default hyperparameters of skip-gram model (Mikolov et al. (2013b)). Note that in the skip-gram model as we scan through the corpus $w_t$ can also be $w_c$'s context word. But in the augmented information here, we do not allow such bi-directional co-occurrence happen since $w_t$ may not be a useful context word for the word $w_c$ in the $(n + 1)$-th domain.

## 6 EXPERIMENTAL RESULTS

We present extensive evaluations to assess the effectiveness of our approach. Following the suggestions of Nayak et al. (2016); Faruqui et al. (2016), we leverage the learned word embeddings as continuous features in several domain-specific down-stream tasks, including document classification, aspect extraction, and sentiment classification. We do not evaluate the learned embeddings directly as in traditional word embedding papers (Mikolov et al. (2013b); Pennington et al. (2014)) because domain-specific dictionaries of similar / non-similar words are in general not available.

### 6.1 DATASETS

We use the Amazon Review datasets He & McAuley (2016) as a huge collection of multiple-domain corpus. We consider each second-level category (the first level is department) as a domain and aggregate all reviews under each category as one domain corpus. This ends up with a rather diverse domain collection. Due to limited computing resources, we limit each domain corpus up to 60 MB. We randomly select 3 domains ("Computer Components", "Cats Supply" and "Kitchen Storage and Organization") as new domains for down-stream tasks on product type classification and sentiment classification. Then we deliberately pick the "Laptops" domain as the new domain for aspect extraction task since the annotation is on Laptop reviews. Each new domain corpus is cut to 10 MB and 30 MB in order to test the practical performance of a small new domain. We randomly select 56 ($m$) domains to train and evaluate the meta-learner. Lastly, three random collections of 50, 100 and 200 ($n - m$) domains corpora are used as past domains.

### 6.2 EVALUATION OF META-LEARNER

We split the 56 domains as 39 ($m_t$) domains for training, 5 domains for validation and 12 domains for testing. So the validation and testing domain corpora have no overlapping with the training domain corpora. This leads to a more general base meta-learner for many unseen new domains.

Table 1: F1 score of fine-tuning on base meta-learner

|  | Cmptr. Components | Kitch. Storage & Org. | Cats Supples | Laptops |
|---|---|---|---|---|
| 10MB | 0.832 | 0.841 | 0.856 | 0.817 |
| 30MB | 0.847 | 0.859 | 0.876 | 0.854 |

We sample 2 ($l$) sub-corpora over the set of reviews from each domain and limit the size of the sub-corpora to 10 MB. We select top 5000 words as word features ($f$). We randomly select 500 words ($|V_{meta}| = 500$) from each domain and ignore words with zero counts on co-occurrence to obtain pairwise examples. This ends up with 80484 training examples, 6234 validation examples, and 20740 testing examples. The f1-score of meta-learner is **81%**.

We further fine-tune the meta-learner for each new domain. We sample 3000 words from each new domain, which ends with slightly fewer than 6000 samples after ignoring zero co-occurrences. We select 3500 examples for training, 500 examples for validation and 2000 examples for testing. The testing f1-score is shown in Table 1. Finally, we empirically set $delta = 0.7$ as the threshold.

### 6.3 DOWN-STREAM TASKS

We use 3 down-stream tasks to evaluate the effectiveness of our approach. For each task, we leverage an embedding layer to store the pre-trained embeddings. We choose our embedding dimensions as 300, which is the same size as many pre-trained embeddings (GloVec.800B (Pennington et al. (2014)) or fastText Wiki English (Bojanowski et al. (2016))). We freeze the embedding layers during training, so the result is less affected by the rest of the model and the training data. To make the performance of all tasks relatively consistent, we leverage the same Bi-LSTM model (Hochreiter & Schmidhuber (1997)) on top of the embedding layer to learn task-specific features from different embeddings. The input size of Bi-LSTM is the same as the embedding layer and the output size is 128. All tasks leverage many-to-one Bi-LSTMs for classification purpose except aspect extraction, which uses many-to-many Bi-LSTM for sequence labeling. In the end, a fully-connected layer and a softmax activation are applied after Bi-LSTM, with the output size specific to each task.

### 6.4 BASELINES

**No Embedding (NE)**: We randomly initialize the word vectors and train the word embedding layer during the training process of each down-stream task. Note that only in this baseline do we allow embeddings trainable.

**fastText**: This is the lower-cased embeddings pre-trained from English Wikipedia using fastText (Bojanowski et al. (2016)). We lower the cases of all corpora of down-stream tasks to match the words in this embedding. Note that although the corpus of Wikipedia contains a wide spectrum of domains covering almost everything of human knowledge, the amount of corpus for a specific domain (e.g, a product) may not be large enough. The total amount of Wikipedia is just several billions of tokens, which is on the same scale as Amazon Review datasets (8 billion tokens).

**GoogleNews**: This is the pre-trained embeddings using word2vec [2] based on part of the Google News datasets, which contains 100 billion words.

**GloVe.Twitter.27B**: This embedding is pre-trained using GloVe (Pennington et al. (2014)) based on Tweets, which have 27 billion words. Note this embedding is lower-cased and has 200 dimensions.

**GloVe.6B**: This is the lower-cased embeddings pre-trained from Wikipedia and Gigaword 5, which has 6 billions of tokens.

**GloVe.840B**: This is the cased embeddings pre-trained from Common Crawl, which has 840 billions of tokens. This embedding corpus is the largest one among all embeddings. It contains almost all web pages available before 2015. We show that although GloVe.840B is general enough on almost any task, its performance is sub-optimal on many domain tasks.

---

[2]https://code.google.com/archive/p/word2vec/

Table 2: Accuracy of different embeddings on product type classification tasks (numbers in parenthesis indicates number of classes)

|  | Cmptr. Cmpnts. (13) | Kitch. Strg. & Org. (17) | Cats Supples (11) |
|---|---|---|---|
| NE | 0.596 | 0.653 | 0.696 |
| fastText | 0.705 | 0.717 | 0.809 |
| GoogleNews | 0.76 | 0.722 | 0.814 |
| GloVe.Twitter.27B | 0.696 | 0.707 | 0.80 |
| GloVe.6B | 0.701 | 0.725 | 0.823 |
| GloVe.840B | 0.803 | 0.758 | 0.855 |
| ND 10M | 0.77011 | 0.74905 | 0.85 |
| ND 30M | 0.794 | 0.766 | 0.87 |
| 200D + ND 30M | 0.793 | 0.759 | 0.859 |
| LL 200D + ND 10M | 0.791 | 0.761 | 0.872 |
| LL 50D + ND 30M | 0.795 | 0.768 | 0.868 |
| LL 100D + ND 30M | 0.803 | 0.773 | 0.874 |
| LL 200D + ND 30M | **0.809** | **0.775** | **0.883** |

**New Domain 10M (ND 10M)**: This is a baseline embedding pre-trained only from the new domain 10 MB corpus. We show that the embeddings trained from a small corpus alone are not good enough.

**New Domain 30M (ND 30M)**: Then we increase the size of the new domain corpus to 30 MB to see the difference affected by the corpus size.

**200 Domains + New Domain 30M (200D + ND 30M)**: Another straightforward embedding is to concatenate the corpora from all past domains and the new domain together to train a mixed embedding. We use this baseline to show that unselected past domain corpora may reduce the performance of down-stream tasks. How to smartly adapt the embeddings into a closer context is crucial.

**Lifelong Past Domains + New Domain (LL [P]D + ND [X]M)** : This is different versions of our proposed method. For example, we use LL 200D + ND 30M to donate embeddings trained from a 30MB new domain corpus and 200 past domains.

## 6.5 PRODUCT TYPE CLASSIFICATION

This task is to classify a review into a product type (leaf-level category in Amazon product category system). There are many product types under each domain (2nd-level category). We use the randomly selected 3 domains as the new domains to form 3 multi-class classification sub-tasks. These domains are: *Computer Components*, *Kitchen Storage and Organization* and *Cats Supplies*. For each sub-task, we randomly draw 1200 reviews for each product type. We drop classes with less than 1200 reviews. This ends up with 13, 17 and 11 classes for *Computer Components*, *Kitchen Storage and Organization* and *Cats Supplies*, respectively. For each sub-task, we keep 10000 reviews as the testing data (to make the result more accurate) and split the rest as 7:1 for training and validation data, respectively. All sub-tasks are evaluated on accuracy. We train and evaluate each sub-task on each baseline 10 times (with different initialization) and average the results.

From Table 2, we can see that the performance of different classification tasks varies a lot. This is mostly caused by the number of classes in each sub-task. The lifelong embeddings LL 200D + ND 30M performs best. The maximum train corpus of this method is in total just 80 MB (30 MB new domain + 50 MB from 200 past domains). The difference in the numbers of LL past domains indicates more past domains have better results. Surprisingly, the GloVe.840B trained on 840 billions of tokens does not perform well enough compared to the limited new domain corpora. But it performs very well on Computer Components, which means this domain is relatively general. Putting all past domain corpora together with the new domain corpus (200D + ND 30M) makes the result worse than to not use the past domains at all (ND 30M). This is because those 200 domains may not be close to the new domains.

Table 3: Performance of different embeddings on aspect extraction task

|  | Precision | Recall | F1-Score |
|---|---|---|---|
| NE | 0.596 | 0.493 | 0.54 |
| fastText | 0.655 | 0.47 | 0.547 |
| GoogleNews | 0.7 | 0.638 | 0.667 |
| GloVe.Twitter.27B | 0.642 | 0.468 | 0.541 |
| GloVe.6B | 0.68 | 0.505 | 0.579 |
| GloVe.840B | 0.722 | 0.6406 | 0.679 |
| ND 10M | 0.663 | 0.57 | 0.613 |
| ND 30M | 0.713 | 0.62 | 0.663 |
| 200D + ND 30M | 0.731 | 0.65 | 0.688 |
| LL 200D + ND 10M | 0.724 | 0.636 | 0.677 |
| LL 50D + ND 30M | 0.736 | 0.637 | 0.683 |
| LL 100D + ND 30M | 0.723 | 0.65 | 0.685 |
| LL 200D + ND 30M | 0.734 | 0.659 | **0.694** |

Table 4: Accuracy of different embeddings on sentiment classification task

|  | Cmptr. Cmpnts. | Kitch. Strg. & Org. | Cats Supples |
|---|---|---|---|
| NE | 0.777 | 0.764 | 0.67 |
| GoogleNews | 0.847 | 0.815 | 0.732 |
| GloVe.Twitter.27B | 0.776 | 0.813 | 0.727 |
| GloVe.840B | 0.877 | 0.859 | 0.779 |
| ND 10M | 0.885 | 0.849 | 0.795 |
| ND 30M | 0.889 | 0.867 | 0.806 |
| 200D + ND 30M | 0.886 | 0.87 | 0.807 |
| LL 200D + ND 10M | 0.882 | 0.85 | 0.773 |
| LL 200D + ND 30M | 0.891 | 0.872 | 0.808 |

## 6.6 ASPECT EXTRACTION

Aspect extraction is an important task in sentiment analysis (Liu (2012; 2015)). We use the dataset from SemEval-2014 Task 4: Aspect-based sentiment analysis (Pontiki et al. (2014)) as a down-stream new domain task. This dataset contains human annotated Laptop aspects and their polarities. It has 3045 training examples and 800 testing examples. We use the Laptop domain corpus from the Amazon Review Dataset as the new domain corpus to train the lifelong embedding. We leverage the original evaluation script to report precision, recall, and F1-score. Again, we average 10 runs of the results.

From Table 3, we can see that aspect extraction is quite different from product type classification. Again, our lifelong embedding performs best. Surprisingly, the performance of 200D + ND 30M is very good. This indicates aspect extraction requires both good general embedding and domain-specific embeddings. For example, good representations of general words can help to identify nearby aspects and good aspects words can also help.

## 6.7 SENTIMENT CLASSIFICATION

We select 6000 4-rating reviews as positive reviews and 6000 2-rating reviews as negative reviews from 3 domains used in product type classification to form 3 sentiment classification sub-tasks. Again, to ensure enough number of valid digits in the results, we use 10000 out of 12000 reviews for testing. The results are averaged over 10 runs.

From Table 4, we can see the performance of most domain-specific baselines is very close (We omit the minor differences between different sizes of past domains). Sentiment classification, in general, requires polarity words to determine the sentiment of a document. Pre-trained general embeddings may introduce non-polarity information into the embeddings. When domain corpus is leveraged, the

Table 5: Performance of Concatenation with GloVe.860B

|  | CC | KSO | CS | Laptops (f1) |
|---|---|---|---|---|
| GloVe.840B&ND 30M | 0.811 | 0.78 | 0.885 | 0.723 |
| GloVe.840B&LL 200D + ND 30M | 0.817 | 0.783 | 0.887 | 0.729 |

difference is small. This is close to our previous experience. A possible explanation is that sentiment classification relies on sentiment words like "good" or "bad". However, those words have similar context words, e.g., "This phone is good." and "This phone is bad.". So the co-occurrence-based training corpus of embedding is not good for learning the embeddings of sentiment words.

### 6.8 FUSING WITH THE OUT-OF-DOMAIN WORLD

Although most cross-domain embedding papers focus on leveraging different existing pre-trained embeddings, we focus on expanding the domain-specific corpus. We believe if we can expand the domain-specific training corpus on a much larger scale (like breaking the training corpus of GloVe.860B into many domains), the performance of the proposed method is much better. However, our focus does not forbid our method from leveraging existing cross-domain transfer learning method (?Bollegala et al. (2015); Yang et al. (2017)). A simple way of leveraging existing embeddings in these papers is to concatenate existing pre-trained embeddings with domain-specific embeddings. To demonstrate our method further improves the domain-specific parts of the downstream tasks, we further evaluate two methods: (1) GloVe.840B&ND 30M, which concatenates new domain alone embeddings with GloVe.860B; (2) GloVe.840B&LL 200D + ND 30M, which concatenates our lifelong embeddings with GloVe.860B.

As shown in Table 5, concatenating embeddings improve the performance a lot. Our method further improves the domain-specific parts of the embeddings. While existing cross-domain embedding methods can only use the 30 MB corpus of the new domain, our method allows those methods to further leverage the expanded corpus.

## 7 CONCLUSIONS

In this paper, we formulate a lifelong word embedding learning process. Given many previous domains and a small new domain corpus, the proposed method can effectively generate new domain embeddings by leveraging a simple but effective algorithm and a meta-learner. The meta-learner is able to provide word context similarity information on domain-level. Such information can help to accumulate new domain-specific training corpus in order to get better embedding. Experimental results show that the proposed method is effective in learning new domain embeddings from a small corpus and past domain knowledge.

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
