# OpenReview forum: "Lifelong Word Embedding via Meta-Learning"
_ICLR.cc/2018/Conference — Reject_

### Official Review · AnonReviewer2 · 2017-11-24
**Addresses an important problem and proposes an interesting approach, but many important modeling details unclear and results only somewhat impressive.**

**Rating:** 5
**Confidence:** 4

**Review:**

Summary:
This paper proposes an approach to learn embeddings in new domains by leveraging the embeddings from other domains in an incremental fashion. The proposed approach will be useful when the new domain does not have enough data available. The baselines chosen are 1). no embeddings 2). generic embeddings from english wiki, common crawl and combining data from previous and new domains. Empirical performance is shown on 3 downstream tasks: Product-type classification, Sentiment Classification and Aspect Extraction. The proposed embeddings just barely beat the baseline on product classification and sentiment classification, but significantly beat them on aspect extraction task.


Comments:

The paper puts itself nicely in context of the previous work and the addressed problem of learning word embeddings for new domain in the absence of enough data is an important one that needs to be addressed. There is reasonable novelty in the proposed method compared to the existing literature. But, I was a little disappointed by the paper as several details of the model were unclear to me and the paper's writing could definitely be improved to make things clearer.

1). In the "Meta-learner" section 4.1, the authors talk about word features (u{_w_{i,j,k}},u{_w_{i,j',k}}). It is unclear what these word features are. Are they one-hot encodings or embeddings or something else? It would really help if the paper gave some expository examples.

2). In Algorithm 1, how do you deal with vocabulary items in the new domain that do not exist in the previous domains i.e. when the intersection of V_i and V_{n+1} is the null set. This is very important because the main appeal of this work is its applicability to new domains with scarce data which have far fewer words and hence the above scenario is more likely to happen.

3). The results in Table 3 are a little confusing. Why do the lifelong word embeddings relatively perform far worse on precision but significantly better on recall compared to the baselines? What is driving those difference in results?

4). Typos: In Section 3, "...is depicted in Figure 1 and Figure 3". I think you mean "Figure 1 and Figure 2" as there is no Figure 3.

---

### Official Review · AnonReviewer3 · 2017-11-26
**Motivation is unclear, detailed descriptions are missing, and technical contributions are limited.**

**Rating:** 3
**Confidence:** 4

**Review:**

In this paper, the authors proposed to learn word embedding for the target domain in the lifelong learning manner. The basic idea is to learn a so-call meter learner to measure similarities of the same words between the target domain and the source domains for help learning word embedding for the target domain with a small corpus.

Overall, the descriptions of the proposed model (Section 3 - Section 5) are hard to follow. This is not because the proposed model is technically difficult to understand. On the contrary, the model is heuristic, and simple, but the descriptions are unclear. Section 3 is supposed to give an overview and high-level introduction of the whole model using the Figure 1, and Figure 2 (not Figure 3 mentioned in text). However, after reading Section 3, I do not catch any useful information about the proposed model expect for knowing that a so-called meta learner is used. Section 4 and  Section 5 are supposed to give details of different components of the proposed model and explain the motivations. However, descriptions in these two sections are very confusing, e.g, many symbols in Algorithm 1 are presented with any descriptions. Moreover, the motivations behind the proposed methods for different components are missing.  Also, a lot of types make the descriptions more difficult to follow, e.g., "may not helpful or even harmful", '"Figure 3", "we show this Section 6", "large size a vocabulary", etc.

Another major concern is that the technical contributions of the proposed model is quite limited. The only technical contributions are (4) and the way to construct the co-occurrence information A. However, such contributions are quite minor, and technically heuristic. Moreover, regarding the aggregation layer in the pairwise network, it is similar to feature engineering. In this case, why not just train a flat classifier, like logistic regression, with rich feature engineering, in stead of using a neural network.

Regarding experiments, one straight-forward baseline is missing. As n domains are supposed to be given in advance before the n+1 domain (target domain) comes, one can use multi-domain learning approaches with ensemble learning techniques to learn word embedding for the target domain. For instance, one can learn n pairwise (1 out of n sources + the target) cross-domain word embedding, and combine them using the similarity between each source and the target as the weight.

---

### Official Review · AnonReviewer1 · 2017-11-27
**This paper studies an interesting problem, but needs improved clarity and experiments.**

**Rating:** 4
**Confidence:** 4

**Review:**

This paper presents a lifelong learning method for learning word embeddings.  Given a new domain of interest, the method leverages previously seen domains in order to hopefully generate better embeddings compared to ones computed over just the new domain, or standard pre-trained embeddings.

The general problem space here -- how to leverage embeddings across several domains in order to improve performance in a given domain -- is important and relevant to ICLR.  However, this submission needs to be improved in terms of clarity and its experiments.

In terms of clarity, the paper has a large number of typos (I list a few at the end of this review) and more significantly, at several points in the paper is hard to tell what exactly was done and why.  When presenting algorithms, starting with an English description of the high-level goal and steps of the algorithm would be helpful.  What are the inputs and outputs of the meta-learner, and how will it be used to obtain embeddings for the new domain?  The paper states the purpose of the meta learning is "to learn a general word context similarity from the first m domains", but I was never sure what this meant.  Further, some of the paper's pseudocode includes unexplained steps like "invert by domain index" and "scanco-occurrence".

In terms of the experiments, the paper is missing some important baselines that would help us understand how well the approach works.  First, besides the GloVe common crawl embeddings used here, there are several other embedding sets (including the other GloVe embeddings released along with the ones used here, and the Google News word2vec embeddings) that should be considered.  Also, the paper never considers concatenations of large pre-trained embedding sets with each other and/or with the new domain corpus -- such concatenations often give a big boost to accuracy, see :
"Think Globally, Embed Locally—Locally Linear Meta-embedding of Words", Bollegala et al., 2017
https://arxiv.org/pdf/1709.06671.pdf

That paper is not peer reviewed to my knowledge so it is not necessary to compare against the new methods introduced there, but their baselines of concatenation of pre-trained embedding sets should be compared against in the submission.

Beyond trying other embeddings, the paper should also compare against simpler combination approaches, including simpler variants of its own approach.  What if we just selected the one past domain that was most similar to the new domain, by some measure?  And how does the performance of the technique depend on the setting of m?  Investigating some of these questions would help us understand how well the approach works and in which settings.

Minor:

Second paragraph, GloVec should be GloVe

"given many domains with uncertain noise for the new domain" -- not clear what "uncertain noise" means, perhaps "uncertain relevance" would be more clear

The text refers to a Figure 3 which does not exist, probably means Figure 2.  I didn't understand the need for both figures, Figure 1 is almost contained within Figure 2

When m is introduced, it would help to say that m < n and justify why dividing the n domains into two chunks (of m and n-m domains) is necessary.

"from the first m domain corpus" -> "from the first m domains"?

"may not helpful" -> "may not be helpful"

"vocabularie" -> "vocabulary"

"system first retrieval" -> "system first retrieves"

COMMENTS ON REVISIONS: I appreciate the authors including the new experiments against concatenation baselines.  The concatenation does fairly comparably to LL in Tables 3&4.  LL wins by a bit more in Table 2.  Given these somewhat close/inconsistent wins, it would help the paper to include an explanation of why and under what conditions the LL approach will outperform concatenation.

---

### Decision · Program_Chairs · 2018-01-29
**ICLR 2018 Conference Acceptance Decision**

**Decision:**

Reject

**Comment:**

While the problem of learning word embeddings for a new domain is important, the proposed method was found to be unclearly presented and missing a number of important baselines. The reviewers found the technical contribution to be of only limited value.